# Fruit and Vegetable Intake Is Associated with Food Knowledge among Children Aged 9–14 Years in Southwestern Ontario, Canada

**DOI:** 10.3390/children9101456

**Published:** 2022-09-23

**Authors:** Louise W. McEachern, Mariam R. Ismail, Jamie A. Seabrook, Jason A. Gilliland

**Affiliations:** 1Human Environments Analysis Laboratory, Western University, London, ON N6A 3K7, Canada; 2Department of Geography and Environment, Western University, London, ON N6A 5C2, Canada; 3School of Health and Rehabilitation Sciences, Western University, London, ON N6A 3K7, Canada; 4School of Food and Nutritional Sciences, Brescia University College, London, ON N6G 1H2, Canada; 5Department of Paediatrics, School of Health Studies, Western University, London, ON N6A 3K7, Canada; 6Department of Epidemiology and Biostatistics, Western University, London, ON N6A 5C1, Canada; 7Children’s Health Research Institute, Lawson Health Research Institute, London, ON N6C 2V5, Canada; 8Schulich School of Medicine and Dentistry, Western University, London, ON N6A 5C1, Canada

**Keywords:** food knowledge, fruit vegetable consumption, young children, diet quality

## Abstract

Interventions to improve dietary quality and intake of fruits and vegetables (FV) among Canadian children have had modest success, and it has been suggested that food knowledge could be key to improvement. Programs have been criticized for insufficiently connecting food knowledge with food skills and decision making about dietary intake. The objective of this study was to investigate factors associated with FV consumption by elementary school children, aged 9–14 years, in Ontario, Canada, including food knowledge, socioeconomic status, sociodemographic characteristics, and the food environment. In 2017–2019, a cross-sectional survey was administered to 2443 students at 60 elementary schools across Southwestern Ontario (SWO), Canada. A parent survey was used to validate self-reported sociodemographic variables. The mean intake of FV reported by these participants was 2.6 (SD 1.1) and 2.4 (SD 1.2) servings/day, respectively. A FV intake below WHO guidelines was reported by 40.7% of respondents. Knowledge score, child age, and parent employment status significantly predicted higher reported intake of FV. This study shows that FV intake among this population group is low, and increased intake is associated with higher food knowledge. To encourage healthy eating, school-based food and nutrition programs that incorporate multiple components and emphasize food literacy are needed.

## 1. Introduction

It is widely accepted that fruits and vegetables (FV) are important for preventing lifestyle-related chronic disease and maintaining overall health [1]. Food-based dietary guidelines and other national recommendations, including Canada’s Food Guide (CFG) 2019, notably encourage the frequent intake of FV [2]. Despite such recommendations, the intake of FV by children in Canada is suboptimal, and evidence suggests that the overall quality of dietary intake of children aged 2–18 years living in Canada is low [3,4].

Poor dietary patterns are linked to an increased risk of diabetes [5], some cancers [6], higher body mass indexes [7], and worse performance in school [8]. Furthermore, FV consumption is positively correlated with well-being in children [9]. Dietary behaviour patterns formed in childhood have been shown to predict lifestyle-related disease in adulthood [10], and therefore, interventions aimed at children and youth have the capacity to influence long-term health.

It is critical to understand the factors associated with children’s dietary intake to address these negative health outcomes. Factors that are well known to influence poor dietary quality among children include the environment, such as availability and access to unhealthy food, individual preferences, and factors such as gender, age, and maternal education [11,12,13]. Attempts to improve dietary quality via interventions that impact these factors have had modest success [14]. Suggested reasons for this are that previous programs have insufficiently connected food knowledge with food skills and decision making about dietary intake, and that the concept known as food literacy could be critical to improving the outcomes of such interventions [15].

‘Food literacy’ is defined as “a set of interconnected attributes organized into the categories of food and nutrition knowledge, skills, self-efficacy/confidence, food decisions, and other ecologic (external) factors such as income security, and the food system” [16]. A relatively new concept, a critical component of food literacy is food knowledge [17]. Despite the importance of food knowledge, a relatively small number of studies have investigated the links between food knowledge and diet [18]. Knowledge about food and nutrition has been shown to correlate with improved dietary intake among adults [18], and among adolescents aged 14 to 19 years, knowledge of how to eat a healthy diet has been identified as one of a number of psychosocial factors (i.e., increased awareness and self-efficacy) associated with healthful dietary behaviours [19]. However, few studies have focused on the relationship between dietary intake and food knowledge in younger children.

As the recommended number of servings of FV, as defined by the CFG, are consumed by fewer than one-third of Canadian children [20,21,22], it has become a federal priority to improve dietary behaviors in children and to develop strategies that improve healthy eating early in childhood [23]. To do so, it is essential to conduct new research to inform public health professionals, policy makers, and educators about factors associated with an inadequate intake of FV. Therefore, the objective of this study was to investigate the factors associated with FV consumption of elementary-school children in Southwestern Ontario (SWO), Canada, including food knowledge, socioeconomic status, sociodemographic characteristics, and the food environment.

## 2. Materials and Methods

### 2.1. Study Design and Population

This cross-sectional study took place in elementary schools across Southwestern Ontario, Canada, during the 2017–2018 and 2018–2019 school years, and included schools in different geographical locations (urban and rural) to recruit students from a range of food environments and socioeconomic characteristics. Sixty schools from two English-language school boards (London District Catholic School Board (LDCSB) and Thames Valley District School Board (TVDSB)), representing the cities of London and St. Thomas, and rural districts within the counties of Middlesex, Oxford, and Elgin, were selected randomly from a list of 160 schools. The principals at selected schools were sent a letter of information (LOI) before agreeing to participate in the study, and a study overview was presented to school staff by the research team. Each participating school was visited by the research team to provide information to children in grades 4 to 8 (aged 9–14 years) to answer any questions, and an LOI, parental consent form, parent/guardian survey, and child assent forms, were sent home. Writen parental consent was required for all students and each provided assent before participating in the study.

### 2.2. Surveys

The parent/guardian survey included questions about individual/family level characteristics and their child’s willingness to try FVs. Questions included those on parent/guardian education level, parental employment status, median family income, family structure, ethnic origin, postal code, and family meal habits. Specifically, each parent was asked the following: In a typical day, about how many servings of fruit do you eat? (Example—1 serving is equal to a piece of fresh fruit, such as an apple, or a small bowl of fruit salad). Each parent was also asked the following: In a typical day, about how many servings of vegetables do you eat? (Example—1 serving is equal to a carrot or other fresh vegetable (do not count French fries or potato chips), a small bowl of green salad, or cooked vegetables). Each parent was asked to estimate how many servings of fruits and vegetables their child eats on a typical day. Response options for dietary variables ranged from 0, 1, 2 up to 4+. The parent/guardian survey took approximately 10–15 min to complete.

The child survey was administered once during the school year by the research team at each participating school, in fall, winter, or spring. The survey was completed by all participating children in a central place (i.e., school gym, or library); the research team reviewed child assent forms, provided instructions, and were available to answer questions during the survey. The self-report child survey included 124 questions under four main topics: sociodemographic information, nutrition and food knowledge, eating habits, and food preferences. Child-reported data were primarily used. However, parent-reported data were substituted for any missing data; child and parent surveys were linked using a unique ID number, which was assigned after surveys were collected and names removed to ensure anonymity. Questions involving individual- and family-level characteristics included sex, age, family structure, and ethnicity. As self-report family income is prone to recall errors and high levels of missing responses, we followed an accepted approach adopted in similar survey-based studies of Canadian schoolchildren of using the median family income (CAD) from the 2016 Census of Canada measured at the dissemination area in which the child’s home was located [24,25,26].

Dietary intake questions were developed by registered dietitians and adapted from previously used surveys [27,28,29]. Dietary behaviors were obtained in part from responses to two multiple-component food frequency questionnaire (FFQ) survey questions. Each student was asked the following: In a typical day, about how many servings of fruit do you eat? (Example—1 serving is equal to a piece of fresh fruit, such as an apple, or a small bowl of fruit salad). Each student was also asked the following: In a typical day, about how many servings of vegetables do you eat? (Example—1 serving is equal to a carrot or other fresh vegetable (do not count French fries, or potato chips), a small bowl of green salad, or cooked vegetables). Response options for dietary variables ranged from 0 up to 4+. A variable for total FV intake was derived from these options. Scores from these two questions were added together and defined as ‘below five’, or ‘five and above’, as suggested by the WHO [30].

Research team members, including registered dietitians and teachers, adapted food knowledge questions from previously used surveys [27,28,29]. A variety of question types were included, such as Likert-type scale, yes/no, multiple choice, true/false, and fill-in the blank. To calculate a total food and nutrition knowledge score, the number of correct responses from 46 questions in the child survey was added. The survey included food and nutrition knowledge questions on the CFG (2007) recommendations, food selection and preparation, healthy eating, locally sourced produce, and nutritional content. For example, “How many servings of FV should children your age eat every day, based on Canada’s Food Guide? (2–8 servings); “Which of the following FV are grown in Ontario? E.g., Apples (True, False)” [31]. The maximum possible score a child could achieve was 46, and the minimum was 0. Data were excluded from score calculations if participants responded to half or less of the knowledge questions (n = 23). All remaining questions that were not responded to were considered ‘I don’t know’ and as a result marked incorrect. A detailed analysis of these questions was carried out as part of another study [31].

### 2.3. Ethics

Research offices of both participating school boards, the school principals, and Western University approved the study protocol.

### 2.4. Data Analysis

Data were analyzed using IBM SPSS Statistics, version 24 (Armonk, NY, USA: IBM Corp). Participant sociodemographic characteristics and total FV intake were explored using descriptive statistics. The direction and strength of the association between continuous sociodemographic variables and FV intake was assessed by the Pearson correlation coefficient. As per Cohen [32], an r ± 0.10, 0.30, and 0.50 were considered a weak, moderate, and strong association, respectively. Group means between categorical variables and FV intake were compared using independent samples *t*-tests. Wherever categorical independent variables had three groups or more, the one-way analysis of variance (ANOVA) was used to compare means of continuous dependent variables. The Tukey post hoc test assessed all pairwise comparisons. The relationship between FV intake and the various predictor variables was examined using multiple regression analysis. To ensure adequate statistical power, there should be a minimum of 10 subjects for every predictor variable of interest [33]. With 5 independent variables in our multiple regression model, this required a sample size of at least 50 subjects with complete data. A *p*-value of <0.05 was considered statistically significant.

## 3. Results

### 3.1. Sample Characteristics

Sociodemographic characteristics of the survey participants are shown in Table 1. Parent/guardian consent was obtained for 25.4% of (2443) the eligible child participants and 2431 child participants gave assent to and completed the child survey. Participants ranged in age from 9 to 14 years with a mean age of 11.2 (SD 1.3); 58.2% of the sample was female. Most participants identified as Caucasian (86.4%) and lived in two parent/guardian households (80.5%). There was a median of two (mean 2.6 (SD 1.1)) children/household, and 35.4% of the sample lived rurally, with a mean distance from the nearest grocery store of 4.0 km (SD 4.0). Of participants’ parents or guardians, more than half (68.8%) had a college- or university-level education, and the mean family income was CAD 87,728.93 per annum (SD 19,234.70).

### 3.2. Dietary Intake and Food Knowledge

The mean servings of fruits reported by these participants was 2.6 (SD 1.1) servings/day and of vegetables was 2.4 (SD 1.2) servings/day. Participants reported a total mean intake of FV of 5.0 servings/day (SD 2.0). A low FV intake was reported by 40.7% of respondents according to WHO guidelines.

The mean ‘total knowledge score’ for the sample was 29.2 points (SD = 7.1) out of a possible 46 (63.5% correct responses) (Table 2).

Associations between children’s reported intake of FVs and various sociodemographic factors are presented in Table 3. Statistically significant differences between gender and reported intake of FV were identified, with females reporting a higher mean intake (servings/day) (5.1, SD = 1.9) compared to males (4.8, SD 2.9); *p* = 0.046. There was a positive relationship between total knowledge score and intake of FV (r = 0.27; *p* < 0.001) and mean family income and intake of FV (r = 0.05; *p* = 0.04). Children’s intake of FV were significantly different by urban/rural location, with mean daily servings of 4.7 (SD 2.2) for city dwellers, 5.0 (1.9) for small city dwellers, 5.1 (2.0) for those who live in a small town, and 5.1 (2.0) for those who live rurally (*p* = 0.04). Higher levels of parental education were associated with increased intake of FV among children (*p* = 0.03). There were no significant differences in the intake of FVs by family structure. No statistically significant associations between intake of FV and ethnicity, school food schedule, or number of children living in the household were found (Table 3).

Multiple regression analysis was used to test whether sociodemographic variables significantly predicted childrens’ intake of FV (Table 4). Regression results showed that the predictors explained 7.7% of the variance (R^2^ = 0.077, F = 14.16, *p* < 0.001). It was found that knowledge score (β = 0.257, *p* < 0.001) significantly predicted higher reported intake of FV, as did child age (β = −0.072 *p* = 0.001) and one parent being employed full time (β = 0.050, *p* = 0.03).

## 4. Discussion

This study describes the intake of FV, and factors associated with higher intake of FV, among a large sample of children in Southwestern Ontario (SWO), Canada. Our results show that the proportions of children who report consumption of ≥5 servings of FVs per day is lower than the national recommendations, and the mean total food knowledge reported by our sample is also quite low at 63.5% [31]. Results further indicate that a higher knowledge score significantly predict a higher reported intake of FVs, and the reported intake of FVs decreases as children became older. Gender, ethnicity, parental socioeconomic status, and urban/rural location did not significantly influence the reported intake of FV.

### 4.1. Dietary Intake

Our results show that just 59.8% of children reported consuming ≥5 servings of FVs per day (as recommended by the WHO; Canada recommends 6 servings/day for this age group), supporting other Canadian studies that indicate the intake of FVs is generally below national recommendations [3]. Encouragingly, the proportion of this sample reporting consumption of ≥5 portions of FVs per day is higher than those reported in other Canadian studies [20]. A likely explanation for this finding is that each school in the study took part in the ‘Ontario Student Nutrition Program’ that provided students with at least one serving (according to the 2007 CFG portion recommendations) of fruits or vegetables three to five days each week. School-based programs are usually well regarded, and targeting the diets of children in this way is an acceptable approach [3]; our results show the importance of developing and investing in such programs.

### 4.2. Food Knowledge

The total food knowledge scores reported by our sample were quite low with an average score of 63.5% (29.2 points out of 46) [31]. Further analyses indicated that, when the influence of urbanicity /rural location, parental education attainment and household income on FV intake was adjusted for, a higher knowledge score significantly predicted a higher reported intake of FVs. To our knowledge, this is the first study that examined the association of food knowledge with intake of total FVs among children aged 9–14. Of the studies that focused on food knowledge and dietary intake, the majority have focused on adults [18], athletes [34,35] and university students [36]. Very few studies internationally have investigated associations between dietary intake and nutrition knowledge among children; those that have been published are among children in Italy, Japan, and the US, and none focused specifically on FV intake. Italian children (N = 445, aged 4–16) were less likely to have two or more snacks daily and to spend more than 3 h in sedentary activities daily if they had high nutrition knowledge scores (OR = 0.89, 95% CI 0.83, 0.97 and OR = 0.92, 95% CI 0.86, 0.99, respectively) [37]. Similarly, children in Japan (N = 1210, aged 6–12) with moderate or high nutrition knowledge showed higher vegetable intake, differing by sex (*p* for trend ≤ 0.0001 for boys and 0.020 for girls in higher grades, and 0.024 for boys and <0.0001 for girls in lower grades) [38]. A study among 532 children aged 11–13 living in urban Ohio, USA, showed a correlation between nutrition knowledge and general food choices for children in grades 7 and 8 (aged 10–13) [39].

Other studies have focused more broadly on food literacy, of which food knowledge is a component [40]. A systematic review assessing food literacy among adolescents examined the effects of interventions to improve food literacy. Overall, this review indicated that greater food literacy was linked to positive effects on dietary behaviors among youth, and while the effect was not strong, many of the included studies had methodological limitations, such as a lack of standardization across dietary intake surveys, or instruments that measure nutrition literacy [40].

Several studies suggest that increasing food knowledge alone may be insufficient to improve dietary intake. A review of 29 studies concluded that, although many interventions showed some positive movement between increased knowledge and markers of diet quality among adults, associations were modest [18]. A review of 31 studies evaluating adherence to nutrition guidance in the US showed that while many participants reported high rates of awareness of national guidance, as well as increased knowledge over time, adherence to the guidance was low [41]. Similar patterns have been described in the UK [42]. Previous research suggests that unreliable program delivery and intensity, along with inadequate program duration are limitations to the success of interventions to increase food knowledge, and that programs should consist of multiple components across the school and home environment [42,43,44]. In addition, many of these studies were conducted among adults and we know that dietary habits established during childhood tend to continue into adulthood [10].

Considered together, these studies suggest that interventions to increase food and nutrition knowledge among children are important and worthwhile, provided that the programs are of sufficient duration and are delivered consistently and reliably [45]. The 2019 Ontario Health and Physical Education curriculum for elementary school children includes a healthy eating component, based on CFG 2019 as part of the healthy living strand. Our study, conducted in 2017–2019, evaluated children’s knowledge based on CFG 2007; it would be worthwhile to assess children’s knowledge of the updated food guidelines.

### 4.3. Age

Children in this study ranged in age from 9 to 14 years, and age was negatively associated with dietary intake—as children increased in age, their reported intake of FV decreased. This finding is perhaps unsurprising, as we know that adolescence is a period where children become more independent and increasingly make autonomous decisions. Social support from friends, family, and schoolteachers are considered essential components in building self-efficacy for decisions relating to dietary behavior [46,47]. Health-related behaviors, such as dietary intake, is an area where adolescents assert their independence [48], and dietary behaviors during this period are marked by an increase in meal skipping, snacking, and fast-food consumption [15]. These results reinforce the importance of promoting healthy dietary behaviors at this stage of life.

### 4.4. Null Findings

This study presents some interesting null findings: gender, ethnicity, mean family income, parental education level, and urban/rural location did not significantly influence the reported intake of FV. A previous in-depth analysis of the food knowledge characteristics of this sample revealed an association between urban/rural settings and food knowledge: participants living in small town and rural settings had higher food and nutrition knowledge scores [31]. However, it would appear that food knowledge is a stronger predictor of FV intake that urban versus rural place of dwelling. Other studies in Canada have shown associations between urban versus rural environment and dietary intake among children and adolescents [49,50], but these studies did not investigate levels of food knowledge among participants, and therefore may have missed the influence of knowledge on dietary intake.

### 4.5. Limitations

Some limitations should be considered when interpreting our study’s findings. We used a cross-sectional design which means that our results are representative of a specific point in time. Parent or guardian consent was acquired for 2443 (25.4%) of the eligible children, and despite the similarity of the sociodemographic characteristics with census results and those of other studies, this relatively low participation rate may affect the generalizability of study results. It is possible that students who participated in this survey may have had different dietary intake habits than their peers, who did not participate in the study, although there is little evidence of this when assessing associations between variables using multivariate statistics [51]. While questions used for dietary intake and food knowledge in the survey had been developed by registered dietitians and used in other studies, the questions were not pretested, or pilot tested. The survey used self-reported measures of FV intake and knowledge, and may be subject to recall bias, particularly in children. However, this may have been mitigated with approaches to reduce the likelihood for recall bias, including a parent survey to validate sociodemographic responses, considerable time to complete the survey, and support for students to ask questions.

## 5. Conclusions

This study shows that FV intake among school-aged children in SWO is low and is associated with food knowledge and age. School-based programs that incorporate multiple components and emphasize the development of food knowledge, therefore, have value among this population. Additional research is advised to assess the knowledge and impact of the CFG 2019 on this population.

## Figures and Tables

**Table 1 children-09-01456-t001:** Sociodemographic characteristics of the sample (N = 2412).

Variable	N *	Proportion (%)	Mean/Median	SD
**Age**	2412		11.2/11	1.3
**Gender**				
Females		58.2		
Males		41.8		
**Grade**	2050			
Five		29.3		
Six		27.5		
Seven		23.5		
Eight		19.8		
**Ethnicity**				
White/Caucasian		86.4		
Other		13.6		
**Number of Children in Main Home**	2216		2.6/2	1.1
**Mean Family Income**	2229		87,728.93/91,264.00	19,234.70
**Urban/Rural**	2224			
City (>100,000 population)		15.1		
Small City (between 10,000 and 100,000)		28.3		
Small Town (>1000 population but <10,000)		21.2		
Rural		35.4		
**Distance to Grocery Store**	2247		4.0/2.12	4.0
<800 m		11.9		
800 m–<1.6 km		28.3		
1.6–<2.4 km		15.6		
2.4–<4.8 km		15.9		
>4.8 km		28.3		
**Maximum Education of Either Parent**	2204			
<High School		7.0		
High School		24.2		
University degree/College		59.8		
Graduate degree		9.0		
**Parent 1 Employment**	2245			
Employed full-time		62.5		
Not employed full-time		37.5		
**Parent 2 Employment**	2185			
Employed full-time		81.5		
Not employed full-time		18.5		

* Any numbers unaccounted for were non-responses.

**Table 2 children-09-01456-t002:** Dietary Intake and Food Knowledge Score.

Variable	N	Proportion	Mean/Median	SD
**Servings Fruit**	2196		2.6/3	1.1
**Servings Vegetables**	2208		2.4/2	1.2
**Servings Fruit and Vegetables**	2189		5.0/5	2.0
**Knowledge Score (out of 46)**	2226		29.2/30	7.1
**Fruit and Vegetable Intake (WHO/median intake)**				
**Low (<5)**	891	40.7		
**High (≥5)**	1298	59.3		

**Table 3 children-09-01456-t003:** Associations between participant sociodemographic variables and servings of fruits and vegetables *.

Variable		Servings FV	Correlation	*p*-Value
**Child age**	**Mean (SD)**	**Mean (SD)**		<0.001
	11.2 (1.3)	2.6 (1.1)	−0.08	
**Gender**		Mean (SD)		0.046
	Female	5.1 (1.9)		
	Male	4.8 (2.1)		
**Ethnicity**		Mean (SD)		0.10
	White (*n = 1784*)	5.0 (2.0)		
	Visible minority (*n = 286*)	4.8 (2.1)		
**Number of Children in the main home**	**Mean (SD)**	**Mean (SD)**		0.81
	2.56 (1.1)	2.6 (1.1)	0.01	
**Mean Family Income**	**Mean (SD)**	**Mean (SD)**		0.04
	87,728.93 (19,234.70)	2.6 (1.1)	0.05	
**Urban/Rural**		**Mean (SD)**		0.04
	City (>100,000 population) (*n = 312*)	4.7 (2.2)		
	Small City (between 10,000 and 100,000) (*n = 570*)	5.0 (1.9)		
	Small Town (>1000 population but <10,000) *(n = 437)*	5.1 (2.0)		
	Rural (*n = 707*)	5.1 (2.0)		
**Distance from grocery store**	**Mean (SD)**	**Mean (SD)**		0.68
	4.0 (4.0)	2.6 (1.1)	−0.01	
**Maximum Household Education**		**Mean (SD)**		0.03
	<High School (*n = 139*)	4.6 (2.0)		
	High School (*n* = 487)	4.9 (2.1)		
	University degree/College *(n = 1186)*	5.0 (1.9)		
	Graduate degree *(n = 178)*	5.2 (2.0)		
**Parent 1 Employment**		**Mean (SD)**		0.04
	Full Time *(n = 1279)*	4.9 (1.9)		
	Not Full Time *(n = 747)*	5.1 (2.1)		
**Parent 2 Employment**		**Mean (SD)**		0.49
	Full Time *(n = 1621)*	5.02 (1.98)		
	Not Full Time *(n = 352)*	4.79 (2.02)		
**Knowledge Score**	Mean (SD)	**Mean (SD)**		<0.001
	29.2 (7.1)	2.6 (1.1)	0.27	

* Any numbers unaccounted for were non-responses.

**Table 4 children-09-01456-t004:** Regression analysis of participant sociodemographic variables, total knowledge score, and total servings of fruits and vegetables (n = 1889).

Variable	B	SE B	β	*p*-Value
**Age**	−0.114	0.035	−0.072	**0.001**
**Mean Family Income**	0.001	0.003	0.013	0.59
**Knowledge Score**	0.074	0.007	0.257	**<0.001**
**Male**	−0.133	0.090	−0.033	0.14
**Caucasian**	−0.125	0.142	−0.021	0.38
**Parent 1 Employed Full-time**	0.189	0.086	0.050	**0.03**
**City**	−0.144	0.151	−0.026	0.34
**Small City**	−0.034	0.112	−0.008	0.76
**Small Town**	0.024	0.120	0.005	0.84
**Education: Less than High School**	−0.113	0.189	−0.014	0.550
**Education: High School**	−0.092	0.106	−0.020	0.389
**Education Graduate**	0.250	0.158	0.036	0.114

Adjusted R^2^ = 0.077.

## Data Availability

Data described in the manuscript, code book, and analytic code will be made available upon request pending ethics approval.

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
