# Peer review of "Fruit and Vegetable Intake Is Associated with Food Knowledge among Children Aged 9–14 Years in Southwestern Ontario, Canada"

_children, 2022, doi:10.3390/children9101456_

Round 1

Reviewer 1 Report

Dear Authors,

Thank you for your manuscript. The paper is interesting and well-written. The topic is important and relevant to public health.

My minor section-specific comments are below.

Please provide the age of the study participants in the abstract.

In the methods section, a more detailed description of the parents' questionnaire would be helpful, providing sample items and the response options. Also, an explanation of how the data from the parents' questionnaires were linked to schoolchildren's questionnaires would make a procedure more clear. Also, information on parents' age would be helpful to provide in Table 1.

Please provide explanations of the abbreviations used in Table 4.

Author Response

Many thanks for your helpful comments.  Please find our response to your review below:

Reviewer 1:

Please provide the age of the study participants in the abstract.

Response: Age added.

In the methods section, a more detailed description of the parents' questionnaire would be helpful, providing sample items and the response options.

Response: We have added a selection of food-related questions from the parent survey (lines 101-108).

Also, an explanation of how the data from the parents' questionnaires were linked to schoolchildren's questionnaires would make a procedure more clear.

Response: We have added the following sentence to clarify how parent and child questionnaires were linked: child and parent surveys were linked using a unique ID number (lines 117-118).

Also, information on parents' age would be helpful to provide in Table 1.

Response: Unfortunately, we did not ask parents to provide their age.

Please provide explanations of the abbreviations used in Table 4.

Response: Thank you for pointing out this oversight – the abbreviations have been removed.

Reviewer 2 Report

Interesting and well-written manuscript.

I read this manuscrip with great interestt. It was well elaborated and presented.

It is based on an original idea, a very actual topic, content with great significance for future studies to be developed, well presented, good citation and recent studies from literature, proper English and redaction.

The introduction provided sufficient background and included all relevant references, recent and relevant ones, also properly cited in the manuscript.

The research methods and design were very well presented, in detail, with strong statistics as well, a good sample chosen, and the target population evaluation. Very good interpretation of the Results, and Discussion at a high level of comparison and citation. The results were presented with important ideas to be followed in future deep studies.

The conclusion is supported by the results and the limitations were quite clearly mentioned.

Community interventions are important to focus on the real needs of the population, adapted to conditions (economics, culture, lifestyle, knowledge, experience), and based on proper research studies, and this is one of this kind.

Author Response

Thank you for your kind comments!